# A Static Sign Language Recognition Method Enhanced with Self-Attention Mechanisms

**DOI:** 10.3390/s24216921

**Published:** 2024-10-29

**Authors:** Yongxin Wang, He Jiang, Yutong Sun, Longqi Xu

**Affiliations:** 1School of Measurement and Control Technology and Communication Engineering, Harbin University of Science and Technology, Harbin 150080, China; 2320600016@stu.hrbust.edu.cn (H.J.); 2220600076@stu.hrbust.edu.cn (L.X.); 2School of Electrical and Electronic Engineering, Harbin University of Science and Technology, Harbin 150080, China; 2303010617@stu.hrbust.edu.cn

**Keywords:** static gesture recognition, high robustness, self-attention enhanced, CNN

## Abstract

For the current wearable devices in the application of cross-diversified user groups, it is common to face the technical difficulties of static sign language recognition accuracy attenuation, weak anti-noise ability, and insufficient system robustness due to the differences in the use of users. This paper proposes a novel static sign language recognition method enhanced by a self-attention mechanism. The key features of sign language gesture classification are highlighted by the weight function, and then the self-attention mechanism is combined to pay more attention to the key features, and the convolutional neural network is used to extract the features and classify them, which realizes the accurate recognition of different types of static sign language under standard gestures and non-standard gestures. Experimental results reveal that the proposed method achieves an average accuracy of 99.52% in the standard static sign language recognition task when tested against the standard 36 static gestures selected within the reference American Sign Language dataset. By imposing random angular bias conditions of ±(0°–9°] and ±(9°–18°], the average recognition rates in this range were 98.63% and 86.33%. These findings indicate that, compared to existing methods, the proposed method not only maintains a high recognition rate for standard static gestures but also exhibits superior noise resistance and robustness, rendering it suitable for static sign language recognition among diverse user populations.

## 1. Introduction

As an indispensable means of communication for people with hearing and speech impairments, sign language has a profound impact on all aspects of their lives [1]. According to the World Health Organization’s Global Report on Health Equity for People with Disabilities in 2022, the total number of people with disabilities in the world’s population of 7.7 billion is 1.3 billion, with hearing and speech impaired people accounting for a significant proportion of the total, highlighting the urgent need for universal access to sign language and assistive technology.

With the leap of Artificial Intelligence (AI) and sensor technology, the integration of wearable devices and human-computer interaction has become the focus of scientific research, which not only broadens the application fields of VR [2] and health monitoring [3], but also shows great prospects in the field of sign language recognition [4]. This technology realizes sign language recognition and conversion by accurately capturing hand movements, opening up a new communication and learning channel for people with disabilities.

In the field of sign language recognition using wearable devices [5,6], researchers have made numerous invaluable contributions. Table 1 summarizes eight different methods for static sign language recognition.

From Table 1, it is evident that gesture recognition technology in the field of machine vision has garnered significant attention due to its convenience, high efficiency, and real-time processing capabilities. Medhanit Y. Alemu’s team [7] employed artificial skin and a Multi-Layer Perceptron (MLP) method, achieving an accuracy of 91.13%. However, this approach is associated with high manufacturing costs and complex data processing. Similarly, Chaithanya Kumar Mammadli’s team [8] utilized Inertial Measurement Unit (IMU) data gloves with MLP, attaining a recognition accuracy of 92%. While these results are promising, they also highlight challenges related to data handling and equipment expenses.

To address the impact of environmental factors, such as lighting and skin color variations, Paolo Sernani’s team [9] explored surface electromyography (sEMG) signals and employed a Long Short-Term Memory (LSTM) network, achieving a recognition rate of 97%. Despite the advantages of sEMG, its complex data processing may hinder its applicability in real-time scenarios.

In contrast to the aforementioned methods, Liufeng Fan and his research team [10] leveraged flexible bending sensors in combination with Convolutional Neural Networks (CNN) and Bidirectional LSTM (BiLSTM), achieving a recognition accuracy of 98.21%. This technique has gained favor among researchers due to its low cost and straightforward data processing.

Furthermore, scholars such as Yinlong Zhu [11] and Weixin Deng [12] attained recognition accuracies of 98.5% and 99.2%, respectively, using BP Neural Networks and Support Vector Machines (SVM). Jungpil Shin’s team [13] utilized a camera with SVM to achieve a recognition accuracy of 99.39%. Meanwhile, C.K.M. LEE’s team [14] employed Leap Motion controllers integrated with Recurrent Neural Networks (RNN), attaining a remarkable recognition rate of 99.44%. However, this method faces limitations in cross-platform applications.

These studies underscore the substantial potential of various technologies in gesture recognition, particularly highlighting the advantages of portable devices in terms of flexibility and real-time processing, which have made them highly regarded in the field of gesture recognition.

Except for this research on recognition accuracy, all of the above methods have not explored in detail the problem of reduced recognition accuracy directly caused by the difficulty of ensuring that each movement achieves the precision of a standard gesture when a person with speech impairment performs a sign language gesture while wearing a wearable device in a real-world application.

On this basis, this paper proposes a static sign language recognition method based on self-attention enhancement, which highlights the key features of sign language gesture classification through the weight function and then combines the self-attention mechanism to assign higher attention to the key features and use a convolutional neural network to extract features and classification. In this paper, the proposed method effectively reduces the low recognition accuracy caused by the unstandardized sign language movement of beginners and then improves the accuracy and robustness of the system.

### Contributions

This paper makes the following contributions to the field of static gesture recognition:Developed a data glove integrated with 5 flex sensors and a 6-axis gyroscope for gesture data collection.Proposed a gesture recognition method capable of identifying 36 static gestures, with two main improvements:A data augmentation method based on noise injection to increase the diversity of the dataset.A feature enhancement technique utilizing a weighted function and self-attention mechanism.Created a dataset of 36 static gestures based on the data collected using the developed glove.

Table 2 presents a comprehensive comparison of the contributions made by the proposed method in this paper against various state-of-the-art approaches introduced by other scholars.

The remainder of this paper is structured as follows: Section 2 explains the overall system architecture, the data collection process, and the composition of the dataset. Section 3 provides a detailed description of the proposed methodology. Section 4 presents and analyzes the experimental results, while Section 5 offers the conclusions of this study.

## 2. Experiment

### 2.1. Overall System Architecture

The overall structure of the gesture analysis system used in this paper is shown in Figure 1, which integrates four key stages of data acquisition, preprocessing, feature extraction, and gesture recognition.

The data collection phase relies on the data glove shown in Figure 2, the technical specifications of which are detailed in Table 3.

In this data collection phase, the Esp32 is used as the main control chip to collect the sensor data of five finger curvatures, noted as input matrix *X*, where the dimension of matrix *X* is 1 row and 5 columns. The data from the inertial sensor, which focuses on comprehensively monitoring the overall attitude of the hand in space, including the information of the angular changes in the *x*, *y*, and *z* axes, is also collected, noted as the attitude matrix A. In order to enhance the comprehensive efficacy and analytical value of the data, this paper implements a data integration strategy to integrate the raw data from the two types of sensors into a data integration, which is consolidated into a matrix *F*, where the matrix *F* can be expressed by Equation (1), where the dimension of matrix *F* is 1 row and 8 columns.
(1)$$F=[X|A]=\left[C_{thumb},C_{index},C_{middle},C_{ring},C_{little},x_{A},y_{A},z_{A}\right]$$

In order to enhance the applicability of the data, this paper further implements the data scaling process. In this process, the maximum value (denoted as *y_max_*) and the minimum value (denoted as *y_min_*) of the scaled data were set, where the input data are denoted as *x*, the minimum value of the input data are denoted as *x_min_*, and the maximum value as *x_max_*. The scaled data range according to the method of Equation (2), which was used to map the raw data of the Flex sensors to the range of 1500–2500, corresponding to the angular values of 0°–180°, to ensure the uniformity of the data and the convenience of analysis.
(2)$$y=y_{min}+\frac{\left(x-x_{min})\times (y_{max}-y_{min}\right)}{x_{max}-x_{min}}$$

### 2.2. Dataset

In this paper, experiments were conducted to construct a new dataset, which is based on the American Sign Language (ASL) dataset [15] shown in Figure 3, and the gesture data were collected with a data glove. The new dataset contains 36 standard static gestures, which not only includes 10 numeric gestures from 0 to 9, but also covers 26 alphabetic gestures from A to Z.

In this experiment, a total of 14,400 samples were collected for 36 gestures during the data acquisition phase. To ensure effective model training, 300 samples for each of the 36 gestures were randomly selected as the training set, totaling 10,800 samples, which covers the diversity of gestures. The remaining 100 samples for each gesture were used as a test set, totaling 3600 samples, to evaluate the model’s performance. The waveforms of these 36 static gestures are presented in Figure A1 of Appendix A.

### 2.3. Data Acquisition Process

This experiment recruited 20 volunteers to collect hand gesture data. Each volunteer was instructed to perform 36 gestures, as shown in Figure 3, and maintain each gesture for 30 s, during which 20 data samples were collected per gesture. To mitigate the effects of muscle fatigue between different gestures, a 10 s rest period was provided between gestures for each volunteer. Additionally, to ensure consistency in the data, an electronic goniometer was used to measure finger angles between gestures. This helped ensure that the same gesture was performed with the same angle across different volunteers, thereby guaranteeing the precision of the standardized gestures.

### 2.4. The Design of Experiments

In order to evaluate the performance of the proposed method in the gesture recognition task, standard gesture experiments and robustness experiments are designed in this paper. The standard gesture experiment aims to evaluate the performance of the proposed method in ideal conditions, while the robustness experiment examines the robustness of the proposed method to non-standard gestures. Table 4 details the hyperparameters of the model constructed by the proposed method. A customized dataset containing 36 standard gestures as shown in Section 2.2 was used for the experiments, and recognition accuracy was used as the evaluation metric. The standard gesture experiments were performed by conducting five independent experiments on the standard gesture test set and calculating the mean value of its recognition accuracy, precision, recall, and F1 score. The robustness experiments, on the other hand, are based on the standard gesture test sets, and four robustness test sets introducing different levels of random angular bias ranges, which are ±(0°–9°], ±(9°–18°], ±(18°–27°], and ±(27°–36°], are constructed to simulate the angular bias in the actual use. The specific construction method is detailed in Section 2.5. For each level of robustness test sets, five experiments were conducted independently, and the mean value of the recognition accuracy, precision, recall, and F1 score were calculated and recorded as a quantitative assessment of the performance of the proposed method within that random angular bias range.

### 2.5. Design of Robustness Test Sets

In order to comprehensively evaluate the robustness of the proposed method, on the basis of the standard gesture test sets, four different levels of robustness test sets are extended by introducing additional random angular bias, namely, four robustness test sets with random angular bias of ±(0°–9°], ±(9°–18°], ±(18°–27°], and ±(27°–36°] to verify the effect of slight to significant bending angle variations that may be encountered during practical applications on the recognition accuracy of the proposed method. As an example, a robustness test set introducing ±(0°–9°] random angular bias was constructed based on Equation (2), and a random integer bias of ±(0–50] was added to each data point in the standard gesture test set to simulate angular deviation in actual use. Random angular bias was added to the standard gesture test sets, but these data were not included in the original standard gesture test set. Instead, they were used to construct a separate robustness test set. Each robustness test set contains 100 gesture samples, totaling 3600 data records, the same size as the standard gesture test set. The remaining three robustness test sets were designed in the same way.

## 3. Method

### 3.1. Overview of the Proposed Method

The proposed method can be subdivided into four core parts: first, the preprocessed matrix *F* is input, the input matrix *F* is noisily injected to form a new matrix, and the formed new matrix is input to the self-attention enhancement module. In this phase, the key features that determine the gesture classification are enhanced by augmenting the weight function of the features to highlight their importance. Subsequently, the self-attention mechanism is utilized to assign higher attention weights to these key features to ensure that the proposed method can focus on the most discriminative features. These features are further extracted and categorized in CNN. Based on the extracted features, gesture classification is performed to achieve accurate recognition of sign language data. This process not only improves the recognition accuracy of the proposed method but also enhances its robustness during use by different users. Figure 4 demonstrates the specific flow of the proposed method.

### 3.2. Data Diversity Enhancement Based on Noise Injection

Deep neural networks require a large number of cases for training. A number of data enhancement strategies have been proposed for image analysis, such as flipping, color space [16], panning, rotation, noise injection [17], image blending, random cropping, and generative adversarial networks [18]. In this paper, noise injection is used to add a small amount of random noise to the input data that is small and consistent with the distribution of the original data, as opposed to the data collected by the data glove.

In order to generate a noise matrix *F_noise_* with the same shape as the input matrix *F*, where the dimension of matrix *F* is 1 row and 8 columns, with a single channel, the matrix *F* is regularized in order to maintain the consistency of the data and scale standardization, and after regularization, normally distributed biased noise obeying a mean of 0 and a standard deviation of 0.001 is added to each element, where the standard deviation is chosen to be 0.001 in order to make sure that the added noise is sufficiently small and will not change the data’s original features [19], and at the same time an effective way to prevent overfitting [20] and increase sample diversity [21]. Then the matrix *F*′ after noise injection can be expressed in the form of Equation (3).
(3)$$F\prime =F+F_{noise}$$

The amount of data in the training set is expanded after noise injection for each static sign language data. The new training set consists of 10,800 data matrices from the original training set together with the added 10,800 data matrices after noise injection, and this process can be expressed by Equation (4).
(4)$$F_{\mathrm{new}}=\left(\begin{array}{l} F\\ F\prime \end{array}\right)$$

As a consequence, the amount of data in the new training set is doubled from the original 10,800 data representing 36 gestures to 21,600 data covering the same 36 gestures. The increase in data volume enhances the diversity of the data and utilizes the limited data more effectively to simulate real-world variations, thus improving the robustness of the model.

### 3.3. Feature Enhancement Based on Self-Attention Mechanism

Self-attention mechanism (SAM) can demonstrate its excellent data processing ability in academic research, which is able to deeply mine and accurately parse the intricate contextual dependencies within the input sequence data [22]. By dynamically adjusting the allocation of attention, the mechanism can intelligently focus on the feature regions carrying key information in the sequence [23], realizing efficient screening and refining of information. And it is widely used in machine translation [24], machine vision [25], and gesture prediction [26].

This paper utilizes the unique advantage of the self-attention mechanism in extracting the core features of the data [27]. Given that the core of the gesture recognition task is to accurately identify the number of straightened fingers that constitute a particular gesture, a threshold range of finger bending angles in the angular interval of 0°–90° is set as the criterion for defining key features. According to Equation (2), the data interval of 2000 to 2500 corresponds to the angular interval of 0°–90°. In this paper, feature enhancement is performed for this feature data interval so as to increase the weight of key features. The feature enhancement process based on the self-attention mechanism can be specifically divided into the following four parts:1.Constructing the weighting function

In order to deeply optimize the data processing and feature representation of the input matrix, a weight function is constructed in this paper. The core purpose of this function is to implement a differentiated weight assignment strategy for each element of the matrix based on the intrinsic importance of the data or its unique attribute characteristics. This process is realized by Equation (5), where *α* = 1.5 and *β* = 0.5. The values of these parameters were determined through multiple experiments conducted within the range of 0 to 2, resulting in optimal performance.
(5)$$f(x_{i})=\left\{\begin{array}{ll} \alpha x_{i} & x_{i}>2000\\ \beta x_{i} & x_{i}\leq 2000 \end{array}\right.$$

2.Feature transformation

After differentially assigning weights to each element in the input sequence by means of a weight function, a further linear transformation operation is performed to generate the three key vectors, namely query (*Q*′), key (*K*′), and value (*V*′) [28]. This process is derived based on Equation (6) and aims to reveal the deep relationships and underlying patterns between data elements through linear mapping.
(6)$$Q^{\prime }=f(x_{i})W^{Q},\;K^{\prime }=f(x_{i})W^{k},\;V^{\prime }=f(x_{i})W^{v}$$
where *W^Q^*, *W^K^*, and *W^V^* are trainable weight matrices.

3.Allocating Attention to Feature Fingers

In order to assign higher attention to the feature fingers mentioned in the previous section during feature extraction to improve the robustness of static sign language recognition, the attention mechanism is applied. The attention mechanism generally relies on calculating the similarity between the input sequences and thus giving different weights to the individual elements, and a key step in calculating the similarity between the elements in the input sequences is to measure the similarity between them by the dot product between the query (*Q*′) and the key (*K*′) generated by the feature transformation in step 2. This mechanism allows the model to automatically focus on the most important parts of the input data, allowing the model to assign higher attention to the feature fingers, a process based on the complete content of the input matrix and aimed at revealing the intrinsic connections between the elements [29].

Specifically, for any two elements *x_i_* and *x_j_* (where *i* and *j* represent the position of the element in the sequence and *i* ≠ *j*) in the sequence, the corresponding query vectors *Q_i_*′ and key vectors *K_j_*′ are first computed by predefined feature transformations, respectively. Subsequently, the dot product operation *Q_i_*′⋅*K_j_*′ is performed, the result of which represents the element-to-element attention score. This score reflects the strength of similarity or correlation between two elements in the feature space, which is realized by Equation (7).
(7)$$E^{\prime }=\frac{Q\prime \cdot K\prime^{T}}{\sqrt{d_{k}}}$$
where *d_k_* is the dimension of the key vector.

The scores are then transformed into probability distributions by the SoftMax function. This is performed by Equation (8).
(8)$${\mathrm{Attention}}^{\prime }=\mathrm{softmax}(E\prime )=\frac{\exp(E\prime )}{\sum \exp(E\prime )}$$

4.Weighted summation

The value vector *V*′ is weighted and summed using the attention weights to generate the final output vector, which is implemented by Equation (9).
(9)$$O^{\prime }=\sum_{j}{\mathrm{Attention}}^{\prime }\cdot V^{\prime }$$

The whole process of the self-attention enhancement module can be represented as the structure shown in Figure 5.

### 3.4. Feature Extraction Based on Convolutional Neural Networks

After the introduction of a self-attention enhancement module to give higher attention to the feature fingers, the subsequent processing flow incorporates the advantages of convolutional neural networks and fully connected layers [30]. Among them, convolutional neural networks are designed to deeply mine and extract key features of finger data enhanced by self-attention mechanisms. This process not only enhances the model’s ability to capture subtle feature differences but also promotes effective aggregation and representation of information. Subsequently, these finely refined features are passed to the fully connected layer (FC), which is responsible for learning and mapping out the complex relationships between the input features and the predefined categories, ultimately realizing the accurate classification of finger data. Figure 6 visualizes the overall architecture of the convolutional neural network.

Table 5 details the key parameters of each layer, including input and output dimensions, convolution kernels, etc., which provide a quantitative basis for the model architecture.

In summary, the proposed method constructs an efficient gesture recognition model architecture by integrating a noise injection module, a self-attention enhancement module, and a convolutional neural network, which not only strengthens the feature extraction capability of finger bending data but also realizes the accurate recognition of sign language gestures through the fine parameter configuration and hierarchical design. In the next section, the proposed method will be systematically evaluated and validated in terms of recognition accuracy and robustness using a rigorous scientific method.

## 4. Results

In this section, the overall experimental results of the proposed method on static gesture recognition are elaborated. Through a series of experimental design and implementation, the performance of the proposed method in terms of feature extraction, recognition accuracy, etc., and also systematically evaluates its noise immunity and robustness in practical application scenarios.

### 4.1. Comprehensive Experimental Performance Evaluation

To ensure the reliability of the experimental results, this paper repeats five experiments independently executed on the training set and the standard gesture test set introduced in Section 2.2. In this paper, the average recognition accuracy is used as the evaluation metric. The average recognition accuracy (*Acc*) is defined as the ratio of the number of correctly classified gesture category samples to the total number of gesture samples [31], and Equation (10) demonstrates its specific calculation method.
(10)$$Acc=\frac{\sum_{1}^{5}\frac{N_{\mathrm{True}}}{N_{\mathrm{Total}}}}{5}$$
where *N_True_* denotes the number of correctly categorized gesture category samples in the test set and *N_Total_* denotes the number of total gesture samples in the standard gesture test set. This metric can provide a quantitative evaluation of the performance of the proposed method on the gesture recognition task. As can be seen from the five experiments and calculating the mean value of the recognition accuracy, the proposed method achieves an average recognition accuracy of 99.52% in the standard gesture experiment.

Figure 7 visualizes the convergence and accuracy curves of both the training set and the standard gesture test set during the training process of the model built by the proposed method. Additionally, it displays the ROC curves of the standard gesture test set, which highlight the trade-off between true positive rate and false positive rate at various thresholds. These curves not only clearly reflect the rapid convergence of the learning process and the steady improvement in recognition accuracy, but also demonstrate the strong discriminative power of the proposed method, as evidenced by the ROC curves showing high true positive rates with low false positive rates. These results strongly confirm the efficient recognition capability of the proposed method on the standard gesture recognition task.

For the robustness experiments, five independent repetitions of each random angular bias test of the same range were conducted separately to calculate the average of the recognition accuracy to ensure the accuracy and consistency of the results. The experimental results show that the average recognition accuracy of the proposed method shows a decreasing trend as the random angular bias range increases. The average recognition accuracy of each robustness test set is 98.63%, 86.33%, 66.72%, and 45.63%, respectively. As shown in Figure 8, the variation of accuracy with different levels of random angular bias applied is demonstrated.

From the analysis in Figure 8, it can be seen that the recognition accuracy decreases with the increase of random angular bias, and the accuracy fluctuation amplitude of multiple trials increases, but the proposed method still maintains the average recognition accuracy of 98.63% and 86.33% in the ranges that the finger bending bias angle is in the range of ±(0°–9°] and ±(9°–18°]. This result further consolidates the high noise immunity and strong robustness demonstrated by the proposed method in practical application scenarios.

### 4.2. Comparison of Identification Methods

In order to verify the effectiveness of the proposed method, the following paper compares the different state-of-the-art methods proposed by scholars based on wearable device gesture recognition, and Table 6 shows the specific comparison results.

By analyzing the data in Table 6, it can be observed that the proposed method achieves significant improvements across all evaluation metrics in the standard gesture recognition task compared to the current state-of-the-art techniques. This demonstrates not only the effectiveness of the proposed method but also highlights its superiority over other approaches. The observed improvements are mainly due to two factors. First, the incorporation of noise injection enhances data diversity, making the model more robust to input variations and improving generalization. Second, the self-attention mechanism allows for more precise extraction of key features by enabling the model to focus on the most relevant aspects of the input data. These factors lead to the enhanced performance of the proposed method.

In addition, in order to fully evaluate the noise immunity and robustness of the proposed methods in this paper, further robustness experiments are conducted for the methods compared above. Table 7 shows the comparative results of the robustness experiments. The experimental results show that the proposed method can still maintain an average recognition accuracy of 98.63% and 86.33% under the introduction of random angular bias conditions of ±(0°–9°] and ±(9°–18°]. These experimental results not only demonstrate the stability of the proposed method in real application scenarios but also further prove the excellent noise immunity and robustness experiments of the proposed method.

### 4.3. Comparison of Ablation Experiments

The core approach of the proposed method to improve robustness relies on two key modules: the noise injection module that increases data diversity through noise injection, and the self-attention enhancement module that applies self-attention mechanisms to strengthen key features. In order to fully assess the specific contribution of these modules to the performance of the proposed method in this paper, a series of ablation experiments will be conducted on the ASL dataset. These experiments will be conducted while keeping other parameters constant, aiming to quantitatively analyze the effects of the noise injection module and the self-attention enhancement module on the performance of the proposed method. After 40 epochs of iterative training on the base convolutional neural network based on adding the noise injection module as well as the self-attention enhancement module, respectively, with the learning rate set to 0.0001, Table 8 demonstrates the specific experimental results.

From Table 8, it can be analyzed that the proposed method improves the recognition rate of standard gestures from 97.33% to 99.52%, which is a 2.19% improvement in accuracy relative to the CNN network without the noise injection module and self-attention enhancement module. After adding the random angular bias of ±(0°–9°], ±(9°–18°], ±(18°–27°], ±(27°–36°], respectively, it can be seen from the comparison of the data in Table 8 that the recognition accuracy increased by 1.32%, 9.31%, 8.91%, and 7.60%, respectively. Figure 9 illustrates the changes in the growth rates.

Finally, in order to further analyze the inference performance of the proposed method, the recognition performance of the proposed method for each gesture category is shown in detail on the standard gesture dataset using the confusion matrix. As shown in Figure 10, the confusion matrix of the gestures is represented. The values on the main diagonal of the confusion matrix indicate the percentage of correctly predicted samples in each gesture category, and the remaining positions indicate the cases where the model incorrectly predicts a given gesture as another category. From Figure 10, it can be observed that the model can recognize 32 gestures accurately, with the most confusion for the numeric gestures 1 and 6 and the alphabetic gestures E and O.

## 5. Conclusions

In this paper, a static sign language recognition method based on self-attention mechanism enhancement is proposed to address the problems of poor noise immunity and low robustness of wearable devices under different usage objects. The method increases the sample diversity through a noise injection module and designs a self-attention enhancement module, which improves the weight of key features in the feature extraction process and enhances the robustness of gesture recognition.

The experimental results show that the proposed method achieves an accuracy of 99.52% under the standard gesture test experiment set of the new dataset customized with reference to the ASL dataset, and the accuracy rate has been improved to varying degrees in comparison with mainstream methods at home and abroad. In the robustness experiments, even under the random angular bias conditions of ±(0°–9°] and ±(9°–18°], the average recognition accuracies can still be maintained at 98.63% and 86.33%, respectively, and the accuracies are still improved compared with the latest mainstream methods, which show excellent noise resistance and robustness.

However, current research mainly focuses on static gesture recognition, while gesture interactions in real applications are usually dynamic and continuous. Therefore, future work will focus on dynamic gesture recognition and work on designing more efficient and accurate real-time interaction models.

## Figures and Tables

**Figure 1 sensors-24-06921-f001:**
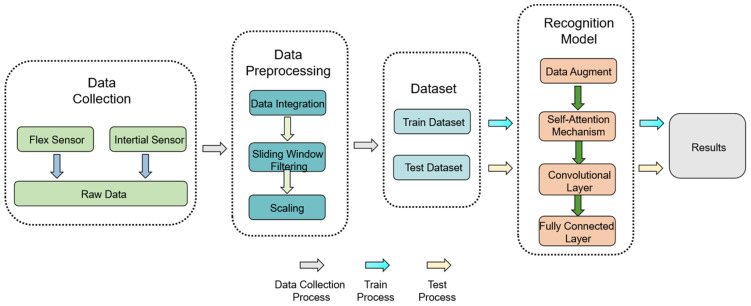
System process flow diagram.

**Figure 2 sensors-24-06921-f002:**
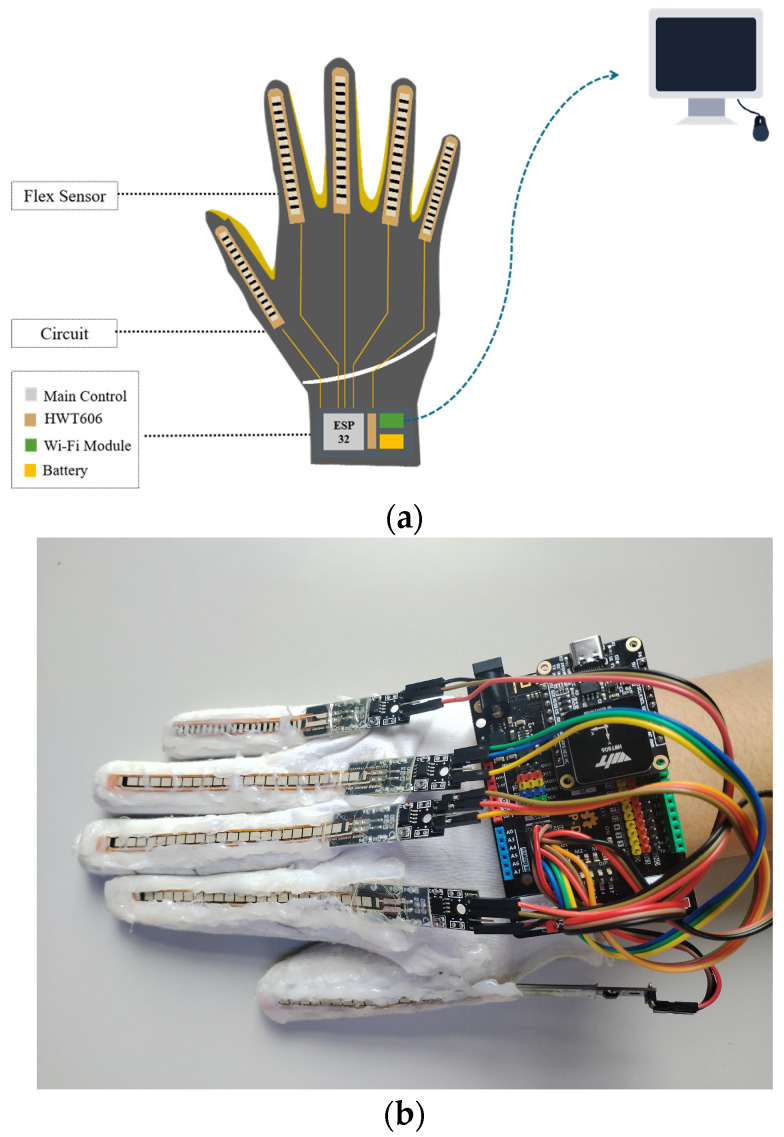
(**a**) Schematic diagram of data glove structure; (**b**) Image of actual data gloves.

**Figure 3 sensors-24-06921-f003:**
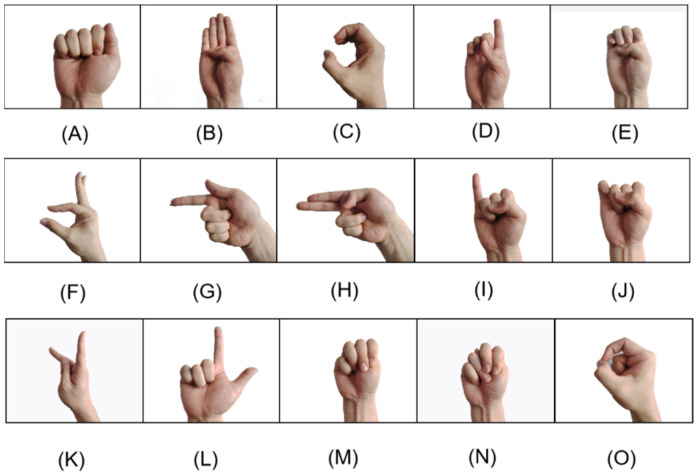
Numerical and alphabetic sign language data set.

**Figure 4 sensors-24-06921-f004:**
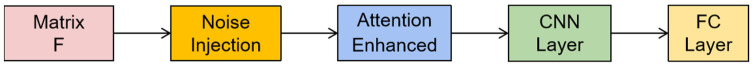
Gesture recognition flow diagram.

**Figure 5 sensors-24-06921-f005:**
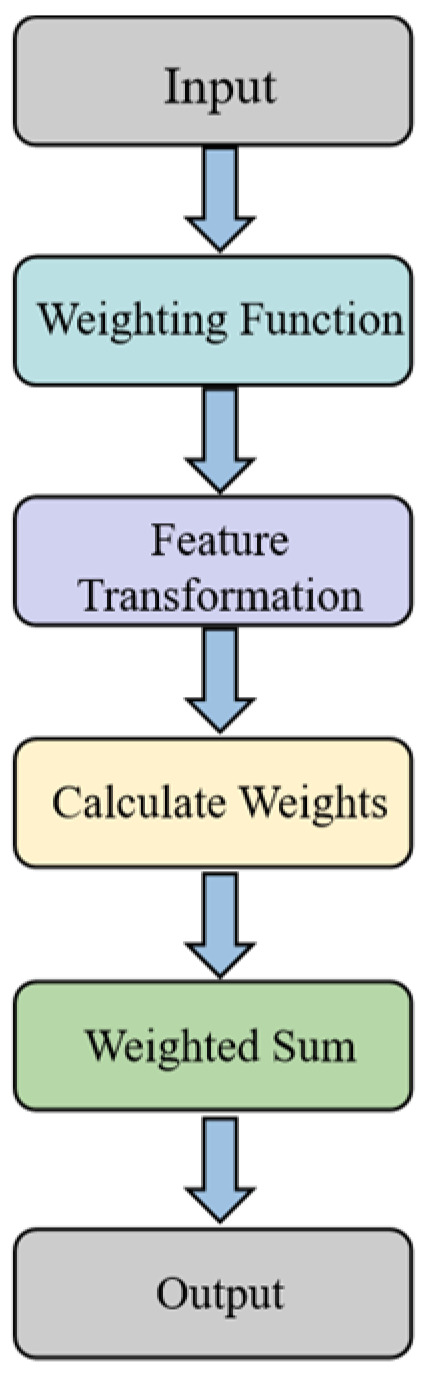
Self-attention enhancement module architecture.

**Figure 6 sensors-24-06921-f006:**
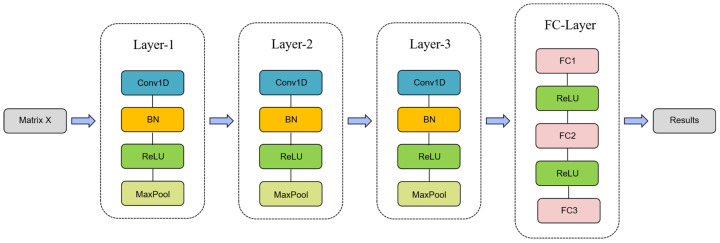
Overall Architecture of Convolutional Neural Networks.

**Figure 7 sensors-24-06921-f007:**
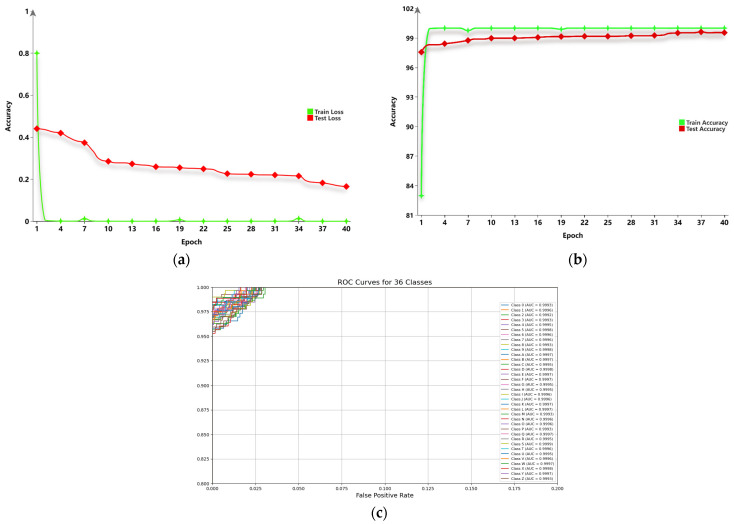
Training set and standard gesture test set: (**a**) Accuracy curves (**b**) Convergence curves. (**c**) ROC curves.

**Figure 8 sensors-24-06921-f008:**
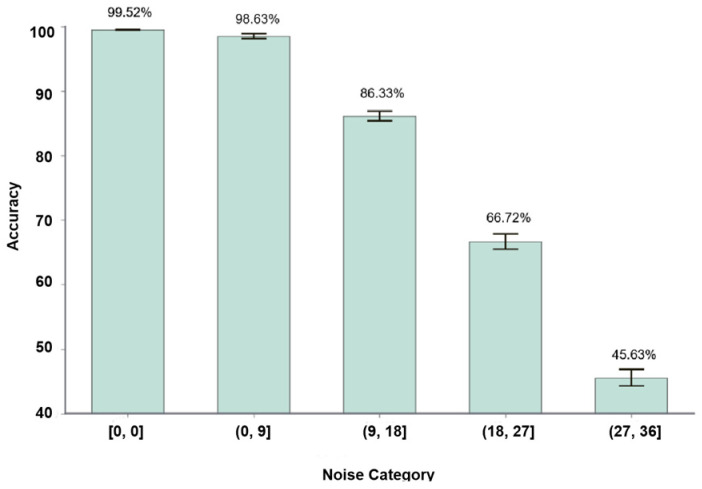
Average recognition accuracy under different levels of random angular bias.

**Figure 9 sensors-24-06921-f009:**
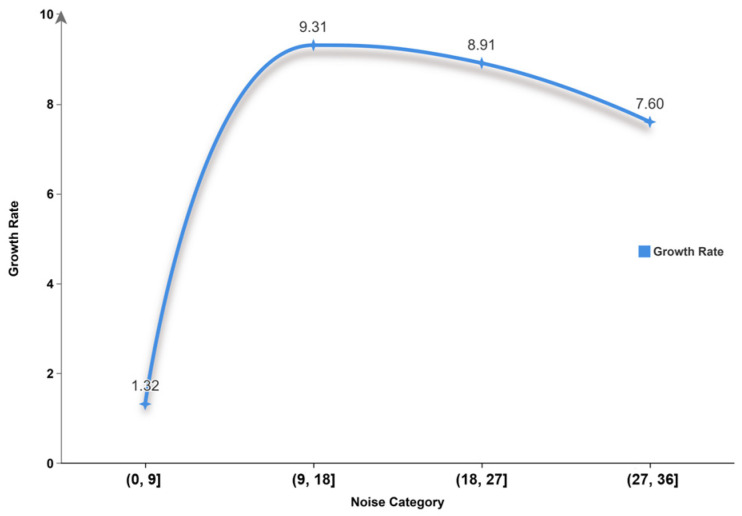
Change in recognition rate growth.

**Figure 10 sensors-24-06921-f010:**
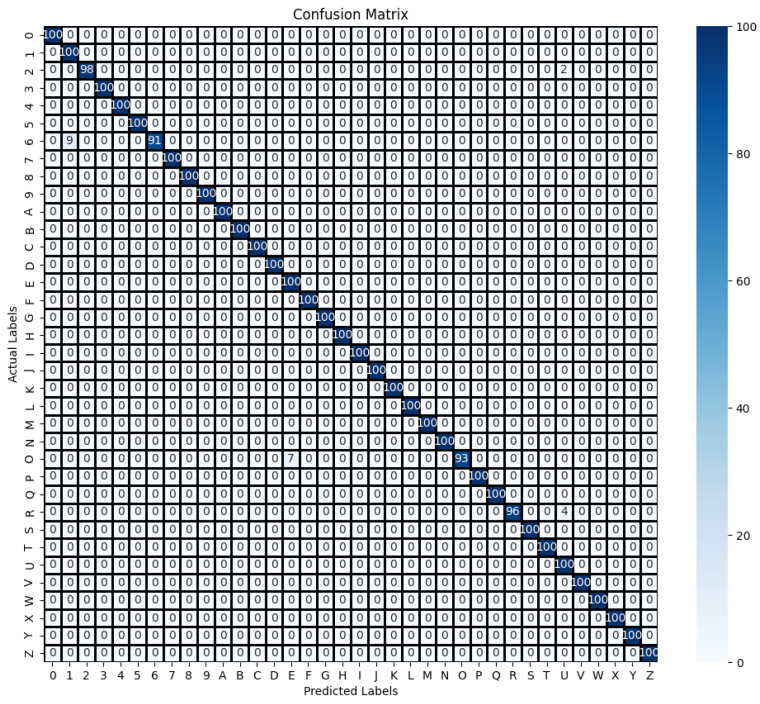
Confusion matrix of standard gesture test set.

**Table 1 sensors-24-06921-t001:** Comparison of different sign language recognition methods.

Researcher	Device	Method	Accuracy
Medhanit Y. Alemu [7]	Artificial Skin	MLP	91.13%
Chaithanya Kumar Mammadli [8]	IMU	MLP	92.00%
Paolo Sernani [9]	sEMG	LSTM	97.00%
Liufeng Fan [10]	Flex Sensor	CNN + BiLSTM	98.21%
Yinlong Zhu [11]	Bending Sensor	BP Network	98.50%
Weixin Deng [12]	Bending Sensor	SVM	99.20%
Jungpil Shin [13]	Camera	SVM	99.39%
C.K.M. LEE [14]	Leap Motion	RNN	99.44%

**Table 2 sensors-24-06921-t002:** Comparison of the Contributions Between the Proposed Method and Existing State-of-the-Art Methods.

Paper	Contributions	Method
[10]	Intelligent Data Glove	Five-Channel Flexible Capacitive Sensor
	Amphibious Recognition Model	AHGR Model
[11]	Intelligent Data Glove	Doped with Carbon Black and Carbon Nanotubes
	Gesture Recognition System	BP Neural Network
[12]	Proposed Method for Guiding Body Movements	Template Matching Method
		SVM
		Kinematic Modeling
Our Method	Proposed Method for Static Sign Language Recognition	Noise Injection
		Self-Attention Mechanism

**Table 3 sensors-24-06921-t003:** Data Glove Details.

Core Components	Details
Chip	ESP32-S3
Bending Sensor	Flex Sensor
Attitude Sensor	HWT606
Data Transfer Method	WIFI

**Table 4 sensors-24-06921-t004:** Hyperparameter Configuration.

Parameter Type	Parameter
Training Size	21,600
Test Size	3600
Training Epochs	40
Optimizer	Adam
Learning Rate	0.0001
Batch Size	150
Loss Function	Cross-Entropy

**Table 5 sensors-24-06921-t005:** Parameter Settings of each layer.

Network	Layer	Input Size	Output Size	Kernel Size	Padding
Conv-Layer-1	Conv1D	1	32	2 × 2	1
	BN	32			
Conv-Layer-2	Conv1D	32	64	2 × 2	1
	BN	64			
Conv-Layer-3	Conv1D	64	128	2 × 2	1
	BN	128			
FC-layer	FC1	128	128		
	FC2	128	64		
	FC3	64	36		

**Table 6 sensors-24-06921-t006:** Comparison of existing advantageous methods.

Number	Method	Accuracy	Precision	Recall	F1-Score	Year
1	CNN + BilSTM [10]	98.21%	98.15%	98.10%	98.12%	2023
2	BP Network [11]	98.5%	98.45%	98.40%	98.42%	2024
3	SVM [12]	99.2%	99.15%	99.10%	99.12%	2024
4	Our Method	99.63%	99.53%	99.47%	99.48%	2024

**Table 7 sensors-24-06921-t007:** Comparison of noise resistance accuracy of different methods.

Angular Bias Range	Method	Accuracy	Precision	Recall	F1-Score
±(0, 9]	CNN + BiLSTM	96.77%	96.70%	96.65%	96.68%
	BP Network	95.17%	95.10%	95.05%	95.07%
	SVM	94.32%	94.25%	94.20%	94.22%
	Our Method	98.63%	98.55%	98.50%	98.52%
±(9, 18]	CNN + BiLSTM	81.65%	81.50%	81.45%	81.47%
	BP Network	80.46%	80.35%	80.30%	80.32%
	SVM	77.13%	77.00%	76.95%	76.97%
	Our Method	86.33%	86.20%	86.15%	86.17%

**Table 8 sensors-24-06921-t008:** Results of ablation experiments.

Angular Bias Range	Experiment	Method	Accuracy
0	Without Noise Injection	CNN	97.33%
		CNN + SAM	98.78%
	Noise Injection	CNN	98.44%
		CNN + SAM	99.52%
(0, 9]	Without Noise Injection	CNN	97.31%
		CNN + SAM	98.33%
	Noise Injection	CNN	97.50%
		CNN + SAM	98.63%
(9, 18]	Without Noise Injection	CNN	77.02%
		CNN + SAM	84.69%
	Noise Injection	CNN	82.33%
		CNN + SAM	86.33%
(18, 27]	Without Noise Injection	CNN	57.81%
		CNN + SAM	65.19%
	Noise Injection	CNN	64.66%
		CNN + SAM	66.72%
(27, 36]	Without Noise Injection	CNN	38.03%
		CNN + SAM	44.64%
	Noise Injection	CNN	42.52%
		CNN + SAM	45.63%

## Data Availability

The dataset is available at https://github.com/jones12138/Dataset, and it was collected on 26 August 2024.

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
