# Peer review of "A Static Sign Language Recognition Method Enhanced with Self-Attention Mechanisms"

_sensors, 2024, doi:10.3390/s24216921_

Round 1
Reviewer 1 Report
Comments and Suggestions for Authors
1. There is no data acquisition process. Additional physical/entity experiments are needed and the detailed description of the data acquisition process should be provided. Specifically, how was the acquisition frequency determined, and what is the duration of each acquisition session? What’s the original dimension of each gesture after data acquisition?
2. Graphs and charts are one of the most important factors in enhancing the persuasiveness of an article; it is recommended to provide the waveform graphs of the data for the gestures.
3. The 36 gesture movements were collected, and each gesture was repeated 400 times. How to ensure the stability and quality of the data acquired during the procedure from being disturbed by hand muscle tremor?
4.In the process of collecting 36 gesture actions, could the repeatability deviation of each gesture be quantified? The deviations caused by the repeated gestures are inevitable. Is it necessary to introduce noise into the data in the following section? It should be explained.
5. Data features are very important in the training process of the algorithm, and what are the features of the original data and the transformed data? Could the authors clarify the characteristics of both the original data and the features after transformation? Additionally, what is the dimension of the data after? What are the feature dimensions (C, H, W) of the input vector of the CNN model?
6. In the manuscript, only one evaluating factor (Accuracy) used to compare the performance of the algorithms, which is not persuasive. Other factors (such as precision, recall, and F1-score) should be also used to compare the efficiency of different methods or models.
Comments on the Quality of English LanguageMinor editing of English language required.
Author Response
Dear Reviewer,
Thank you for your valuable feedback on my manuscript. I have provided point-to-point responses to each of your comments in the attached document.
I appreciate your time and effort in reviewing my work.
Best regards,
He Jiang
School of Measurement and Control Technology and Communication Engineering, Harbin University of Science and Technology, Harbin City, Heilongjiang Province

Reviewer 2 Report
Comments and Suggestions for Authors
Aiming at the problem of static sign language recognition, the authors present a method for analysis of data derived from data gloves. Their method employs a self-attention mechanism in order to assign the higher weight to key features in the feature extraction process performed by convolutional neural network. Thus the recognition rate has been slightly improved. The reviewer would describe the novelty/originality as average but - generally - the manuscript is well-written, the method is presented in detail, its robustness is confirmed by experiments. References are satisfactory.
Some remarks regarding the manuscript are following:
- Page 1, lines 13-15: the same sentence (in slightly different form) is repeated twice
- Page 3, line 102: Equation 1 - index "f" is not explained
- Page 3, line 109: Equation 2 - the meaning of "x" is not explained
- Page 7, line 225: Equation 5 - On which basis were the coefficients alpha and beta determined? Why are their values alpha=1,5 and beta=0,5?
- Page 7, lines 235-237: The sentence is unclear (predicate is missing)
Comments on the Quality of English LanguageMinor improvements required (described in "Comments and suggestions for authors")
Author Response

(The authors gave the same response as above.)

Reviewer 3 Report
Comments and Suggestions for Authors
The authors present the article entitled “A Static Sign Language Recognition Method Enhanced with Self-Attention Mechanisms”
The article presents the following concerns:
-
Add hyperlinks to tables, figures, and references.
-
Avoid using first-person sentences. Use third-person or passive voice sentences in all manuscript.
-
Please check that all acronyms are defined in the main text. For example, define AI on line 36.
-
Lines 48-71: Please cite works according to the author's guidelines.
-
I recommend mentioning the main contributions of the work and describe the structure of the text at the end of the introduction.
-
Lines 50-52: The authors mention a 2018 work where knn is used for gesture detection and its disadvantages such as background and skin color. However, in lines 68-71, the controversy that exists with portable devices is mentioned. It is recommended that the authors describe how these devices are recommended despite being invasive, compared to non-invasive ones such as image processing techniques. It is recommended that they review more recent works such as these, for example: https://doi.org/10.3390/s21175856, where they use RGB images for American Sign Language using CNN in order to highlight the novelty of the work.
-
Subsection 2.1: I suggest that the authors described how they validated that the data obtained from the dataset were acquired correctly.
-
I suggest to add a table that compares the main contributions of the work vs the already reported in the state-of-the-art.
-
Figures must be vectorized in order to see details.
-
year of references 3, 16, and 20 are missing
-
Avoid putting section titles followed by subsection titles.
-
Line 29-30 can be justified with the work: MediaPipe Frame and Convolutional Neural Networks-Based Fingerspelling Detection in Mexican Sign Language.
-
There are several typos, please correct. For example, in line 113 (dataset[14]), there should be a space between the reference and the word.
-
Specify the number of volunteers for the creation of the database. Also the sampling time and the duration of the tests. In general, it is necessary to specify in a detailed and clear way the methodology that was carried out for the acquisition of the database.
-
How was the selection of the hyperparameters carried out, including activation functions, number of layers and number of neurons? Please specify.
-
Specify the Self-attention mechanism used.
-
It is recommended to use other metrics such as F1 score, sensitivity, specificity or AUC for the evolution of the model
The following misspellings should be checked:
-
line 72: “On this basis, in this paper we proposes a…” should be rewritten by “This paper proposes…”
-
line 157: “test set, but were constructed…” should be rewritten by “ set. Still, they were were constructed…”
Author Response

(The authors gave the same response as above.)

Reviewer 4 Report
Comments and Suggestions for Authors
1. In Tabel 1, presenting data sorted by accuracy would be better.
2. Figure 1 and 2, figure quality is poor. The font size of the figures should be clear and easy to read. Check it again.
3. According to Table 5 data, the proposed method's performance is slightly improved than other methods. the authors need a proper explanation of why the proposed method performs better than others. Describle the key contribution.
4. In the result and discussion section, authors should provide a RoC curve.
5. Authors should try to compare the proposed method with other methods with distinct labels for noisy data.
Comments on the Quality of English LanguageThis paper needs "English proof-reading". Editing is necessary, but not much.
Author Response

(The authors gave the same response as above.)

Round 2
Reviewer 1 Report
Comments and Suggestions for Authors
Minor editing of English language required.
Comments on the Quality of English LanguageMinor editing of English language required.
Reviewer 3 Report
Comments and Suggestions for Authors
the manuscript can be accepted for publication